# Baking Quality Assessment of Twenty Whole Grain Oat Cultivar Samples

**DOI:** 10.3390/foods10102461

**Published:** 2021-10-15

**Authors:** Saara Sammalisto, Miikka Laitinen, Tuula Sontag-Strohm

**Affiliations:** Department of Food and Nutrition, Faculty of Agriculture and Forestry, University of Helsinki, Agnes Sjöbergin katu 2, P.O. Box 66, FI-00014 Helsinki, Finland; miikka.laitinen@helsinki.fi (M.L.); tuula.sontag-strohm@helsinki.fi (T.S.-S.)

**Keywords:** oat, whole grain oat, oat cultivars, oat bread, oat baking, dough yield, optimisation

## Abstract

Whole grain oat has become an increasingly popular baking ingredient. Still, oat baking poses many industrial challenges because the baking quality criteria have not been set for whole grain oat flours, and cultivar variation remains unknown. We aimed to assess the baking quality variation of twenty whole grain oat cultivar samples, and to identify the factors that caused the variation. It was hypothesised that by optimising the water absorption of the dough (i.e., dough yield) by test baking method, the best baking potential could be achieved for all oat cultivar samples. The baking trials were conducted as whole oat baking, without wheat or gluten additions. In most of the samples, good baking quality was obtained by dough yield optimisation. The highest specific volumes (1.9–1.93 mL/g) and best crumb properties were achieved in the samples with the highest optimal dough yields, 205. However, baking quality varied, as all samples could not be baked with good quality at high dough yields. Additionally, small median particle size and high fat content of the oat flours were related to good baking properties of whole grain oat at optimised dough yield (*p* < 0.05). These findings can benefit the development and the optimisation of industrial oat baking processes.

## 1. Introduction

Whole grain oat has gained great popularity as a healthy baking ingredient. Whole grain oat is rich in soluble fibre, β-glucan, and for example, the average β-glucan content was about 4% in oat cultivars grown in Finland [1]. Oat β-glucan has several health claims approved by EFSA [2] and FDA [3], which are related to cardiovascular health. Today, oat breads with 100% of the grain ingredients from oat are commercially available but still, whole grain oat baking is challenging because the structure-forming gluten is missing from the dough, and oat cultivar variation in baking quality remains unknown. The traditional uses of oat have been oat flakes and oat porridge, and mixed wheat-oat breads, where oat has been a minor ingredient in baking. No baking quality criteria have been elucidated for whole grain oat flours, which cause challenges in industrial oat baking. In wheat, the baking quality is influenced by wheat genotype, growing environment, and their interaction. Likewise, oat flake properties are influenced by the growing environment and oat cultivar, which also cause variation in oat porridge quality [4,5,6]. Recently, great variation was reported in the physicochemical oat grain and groat properties of 30 oat cultivar samples grown in Finland [1]. Thus, it is reasonable to hypothesise that also the baking quality varies among the oat cultivar samples. This variation needs to be investigated to identify the factors that cause the variation, and to achieve the full baking potential of whole grain oat.

Lack of gluten and high amount of β-glucan make whole grain oat special among the bread-making cereals, and oat baking differs largely from wheat baking. Wheat gluten is capable to produce a dough that retains gas and produces an aerated bread structure, while non-wheat breads tend to remain dense and low in volume [7]. In wheat dough, optimal water absorption is crucial in obtaining the best baking quality, and wheat dough rheology is sensitive to the water content [8]. Optimal water absorption of the dough is essential in gluten-free baking, as well [9,10]. From a technological perspective, whole oat baking can be considered baking without gluten, as oat proteins do not form a viscoelastic network and dough-like structure as gluten proteins do. Oat baking technology has been developed based on rye baking and gluten-free baking, regarding dough stickiness and dough fortification with hydrocolloids. High dough yields (i.e., water absorptions of the doughs) are typical in gluten-free baking. Hydrocolloids are added to the doughs to form a gel network and create a stable structure [7].

Whole grain oat baking has got little attention in the literature, since most oat baking studies have been conducted as mixed oat-wheat baking [11,12]. It has been reported that even a 10% addition of oat bran to the wheat dough can decrease the loaf volume [13]. Oat processing includes a heat treatment (kilning), and sufficient treatment is crucial to obtain a good baking quality with oat [12]. According to our knowledge, no studies have been published about whole grain oat baking where kilning and milling of oat cultivar samples have been controlled.

The objective of this study was to assess the variation in the baking quality of twenty whole grain oat cultivar samples that were kilned and milled in similar conditions. The oat samples were grown in Finland in 2018–2019. It was hypothesised that the baking quality of oat differs among oat cultivar samples, as their physicochemical properties differed. We aimed to assess the range of the variation in baking quality, and to identify the factors that caused the variation. Baking qualities were evaluated after dough yield optimisation by test baking method, so that the optimal baking quality could be achieved in all oat cultivar samples.

## 2. Materials and Methods

### 2.1. Materials

Whole grain oat flours representing 20 oat cultivar samples were studied, and the samples were obtained from Boreal Plant Breeding Ltd., Jokioinen, Finland; Peltosiemen Ltd., Forssa, Finland; Vääksyn Mylly Ltd., Vääksy, Finland; Plantanova Ltd., Ruukki, Finland, Raisio plc, Raisio, Finland, and Lantmännen Agro Ltd., Vantaa, Finland. This study was a part of a larger study, the OatHow project. The basic chemical properties of the oat cultivar samples were published previously in the project [1], and consistent oat sample codes are used in our study. The studied oat cultivar samples (F11–30) were grown in Finland in 2019 except for one sample, which was grown in 2018. Oat groats were dehulled, heat-treated (kilned), flaked, and milled on an industrial scale by Vääksyn Mylly Ltd., (Asikkala, Finland). For the baking trials, table salt (Meira Ltd., Helsinki, Finland), syrup (DanSukker, Suomen Sokeri Ltd., Kantvik, Finland), baker’s yeast (Suomen Hiiva Ltd., Lahti, Finland), and psyllium (Finax Finland Ltd., Lohja, Finland) were purchased from the local supermarket in Helsinki, Finland.

### 2.2. Methods

#### 2.2.1. Oat Flour Analyses

The chemical compositions of the oat cultivar samples were published previously in this project [1]. For this paper, the particle size distributions of the oat flour samples were analysed with Mastersizer 3000 (Aero S, Malvern Instruments, Malvern, UK), with a refractive index of 1.47. Volume-based median particle diameters (D_50_), the means of the particle diameters (D_4,3_), and the particle diameters, that 90% of the particles are smaller (D_90_) were measured. Moreover, the moisture contents (AACC 44–15A, [14]) and centrifugal water holding capacities (WHC) (AACC 56–30, [15]) were analysed from the oat flour samples, the latter with small modifications. In the centrifugal WHC analysis, the samples were incubated at room temperature for 20 min between mixing and centrifuging the samples for better reproducibility of the results. In the original method, the incubation was not specified. All oat flour analyses of this study were conducted in triplicate.

#### 2.2.2. Dough Yield Optimisation

Dough yields (i.e., water absorptions of the doughs) were optimised by test baking method separately for each oat cultivar sample. In test baking trials, the straight dough baking process was developed and monitored. The dough size of 400 g for individual loaf bread was selected, as the bread size was suitable to evaluate the crumb properties, such as porosity and bread staling during storage, and the doughs did not exceed the baking pans during proofing and baking. The same proofing and baking times were suitable for all the samples, as the bread structures did not collapse during proofing, and the bread crumbs of all cultivar samples were mature after baking, without burning of the crust. It was observed that dough yield optimisation had a major influence on the baking quality. Thus, similar proofing and baking conditions were used for all the samples. Dough yields were considered optimised for the oat cultivar sample when the dough was workable by hand, the overall appearance of the bread was good, the crumb had even porosity, and there was no detectable rawness in the bread crumb.

The straight dough baking recipe contained whole grain oat flours (at 14% moisture content), an optimised amount of tap water (85–105% of flour basis, fb), syrup (4% of fb), baker’s yeast (3% of fb), salt (2% of fb), and psyllium (2% of fb), and the total weight of the dough batch was 1.5 kg in test baking trials. Psyllium was mixed with half of the dough water and left to set for 10 min. Then, all ingredients were mixed at low speed (100 rpm, 2 min) and then at high speed (300 rpm, 5 min) with a mixer with a paddle attachment (Metos, Instrumentarium Ltd., Finland). The dough rested for 15 min at room temperature before baking. Dough pieces of 400 g were baked by hand, placed in oiled baking pans (18 cm × 6 cm × 8 cm), and proofed at 35 °C (100% RH) for 30 min (Lillnord TopLine, Odder, Denmark). The breads were baked in a convection oven at 205 °C for 30 min, with 20 s of steaming at the beginning (Sveba Dahlen, Fristad, Sweden). The breads were allowed to cool down for 60 min before evaluation.

#### 2.2.3. Baking Trials with Storage Tests

To evaluate the baking quality and storage stability of each sample, the oat cultivar samples were baked at optimised dough yield in three replicate doughs. Apart from an increase in dough size (from 1.5 kg to 5 kg), the recipe and the baking process followed the same practice as in test baking trials. The proofing and baking conditions were monitored at increased dough size, and it was observed that the process remained suitable. After 60 min from baking, the breads were weighed to measure the bake loss. Specific volumes of the breads were measured with a laser scan, VolScan Profiler (VSP300, Stable Micro Systems, Godalming, UK).

Bread qualities were evaluated by two oat baking expert assessors, who conducted the baking trials, and they were the two first authors of this study. In the test baking method, the baking quality evaluation includes instrumentally measured bread quality (such as bread hardness, specific volume) but additionally, it includes the test baker’s expert assessment of the quality parameters and quality defects that cannot be measured instrumentally, such as unevenness and rawness of the bread crumb (detected visually), taste faults (bitterness), and mouthfeel faults (hardness or stickiness). In test baking, the quality can be assessed properly only when both these aspects are considered.

Bread quality evaluation by the expert assessors was based on fulfilling the set quality criteria and detection of the quality defects in the breads. Bread crumb structure (evenness of the porosity), bread shape, taste, and mouthfeel were rated according to set quality ratings (rates between 1–3 for each quality parameter; 3 excellent, 2 good, 1 satisfactory). Bread crumb structure was rated excellent (3) if the crumb had an aerated structure and even porosity (Figure 1A). Single irregularities in the porosity were accepted since the crumb structure in whole grain oat baking is not comparable to wheat baking. Crumb structures were rated good (2) if some more irregularities or dense areas were observed, but the porosity was still rather even, while in satisfactory breads (1), more irregularities were observed (Figure 1A). Bread shape was rated excellent (3) if the crust shape was rounded like in wheat baking, good (2) if the shape was flat but still uniform, and satisfactory (1) if the crust surface had an irregular shape (Figure 1B). Excellent taste (3) was good and neutral, without off-flavours. The taste was rated good (2) if a minor off-flavour was observed, and satisfactory (1) if the off-flavor was notable. Off-flavour was mainly caused by bitterness. The mouthfeel was rated excellent (3) if the mouthfeel was soft and easily chewed, good (2) if some hardness or stickiness was observed, and satisfactory (1) if the bread was hard to chew or the mouthfeel was sticky.

Bread crumb texture was analysed with a two-bite compression test (Texture Profile Analysis, TA.XT2i Texture Analyser, Stable Micro Systems, Godalming, UK) after 1, 2, and 3 days of storage at room temperature. A 5 kg load cell and a cylindrical probe with a diameter of 36 mm were used in the analysis. Four 25 mm thick slices were cut from each replicate bread, and the crust was removed carefully with a knife. The bread crumb sample was placed under the probe and compressed twice into 40% deformation at 5 mm/s speed, with a 5 s rest between the compressions. Bread crumb hardness was measured as the peak force (N) during the first compression. Staling rate was measured as the increase of crumb hardness (N) between storage days 1 and 3.

Dough consistency was measured from the oat doughs at optimized dough yield with backward extrusion method, with a 5 kg load cell (TA.XT2i Texture Analyser, Stable Micro Systems, Godalming, UK). Five replicate dough samples were compressed into 50% deformation with a disc probe (diameter of 35 mm) at 1.5 mm/s speed in a cylindrical container (diameter of 50 mm). The container was filled with the dough sample to a height of 40 mm, and the sample surface was oiled to prevent sample adherence to the probe. The dough samples were prepared for the analyses similarly to the test baking trials, but without yeast. Dough consistency (N∙s) was measured as the area between the force and the time axes during compression.

#### 2.2.4. Statistical Analyses

In the tables and figures, the average values of the measurements are presented. All analyses were performed at least in triplicate. The error bars and error values represent standard errors of the means (SEM). Correlation analysis was conducted with Pearson’s correlation coefficient (IBM SPSS Statistics 27, IBM, USA). Analysis of variance (ANOVA) was done with Tukey’s HSD test (IBM SPSS Statistics 27, IBM, USA). Statistical analyses were conducted at a significance level of *p* < 0.05.

## 3. Results

### 3.1. Oat Flour Analyses

Particle sizes and centrifugal WHC of the flours varied greatly between the oat cultivar samples (Table 1). Median particle sizes (D_50_) varied between 112–204 μm, average particle diameters (D_4,3_) between 229–316 μm, and D_90_ values between 606–826 μm. WHC of the oat flour samples varied between 1.08–1.45 mL/g.

### 3.2. Baking Trials

#### 3.2.1. Dough Yield Optimisation

Centrifugal WHC of the oat flour samples were measured to predict the optimal dough yield in test baking trials, but it was observed that these properties were not related to each other. This was also confirmed in correlation analysis (Section 3.3). Thus, the oat flour samples were test baked for dough yield optimisation.

The importance of the dough yield optimisation was clearly demonstrated as in most of the samples, good baking quality was obtained by dough yield optimisation. However, oat bread samples varied greatly regarding their overall quality, crumb structure, and bread shape (Figure 2A–D). High optimal dough yield was beneficial in oat baking, as the best baking qualities were obtained at high dough yields (Figure 2A). Some samples could not be baked at high dough yields with good quality, as the dough was too sticky to handle or the crumb structure had impaired quality, but the baking quality improved at lower dough yields (Figure 2B). Despite the dough yield optimisation, the crumb structure remained dense in some samples (Figure 2C). Among the 20 samples, one cultivar sample (F23) had an unusual baking behaviour, as the baking quality remained unsatisfactory despite optimising the dough yield (Figure 2D). Between dough yields 185–205, the dough was sticky, and the crumb was torn, crumbly, and dry. The sample was baked at a dough yield of 185, as the dough was easier to handle compared to higher dough yields, although the crumb structure remained torn and unsatisfactory.

#### 3.2.2. Baking Quality at Optimised Dough Yields

In baking trials with storage tests, oat cultivar samples were baked at their optimised dough yields, which varied between 185–205 (Table 2). Great variation was observed in the baking quality parameters, as dough consistencies, bake losses, and specific volumes showed statistically significant (*p* < 0.05) variation between the samples (Table 2). Dough consistencies varied between 135–359 N∙s, bake losses differed between 14.9–17.6%, and the specific volumes of the oat breads varied between 1.45–1.93 mL/g (Table 2).

Also, bread crumb hardness showed notable variation between the samples (Figure 3). After one day of storage, crumb hardness varied between 19–40 N (27 N on average), while after three days of storage, crumb hardness varied between 20–46 N (31 N on average). Staling rate (the increase of crumb hardness between storage days one and three) ranged between 0–8.4 N, and it was 3.6 N on average (Table 2). In 30% of the samples, bread crumb hardness did not increase significantly during storage, and therefore, the staling rate was 0.

#### 3.2.3. Quality Evaluation of the Oat Breads

The oat bread samples showed variation regarding their crumb structure, bread shape, taste, and mouthfeel (Appendix A, Table A1). The crumb structure was good or excellent in most of the oat bread samples (65%) (Figure 1A in Section 2.2.3). A good oat bread shape was more challenging to obtain, as 40% of the breads were rated good or excellent (Figure 1B in Section 2.2.3). The taste of the breads was mainly rated good and excellent (80%), likewise the mouthfeel (60%). As these four quality ratings were summed together, 25% of the samples had excellent grades (3) on average, while most of the samples (65%) were rated good (2) on average, and a minority of the samples (10%) had satisfactory grades (1) on average. Thus, most of the oat cultivar samples were baked with good or excellent quality.

### 3.3. Statistical Analyses

Pearson’s correlation analysis was conducted from the baking quality data and physicochemical data of the oat flour samples (Table 3). The analysis was done from the 19 satisfactory samples (excluding sample F23), and at a significance level of *p* < 0.05.

High dough yield correlated with low dough consistency (*p* < 0.001), as expected, but also with high bake loss (*p* < 0.001), high specific volume (*p* < 0.001), and soft crumb structure (*p* < 0.01). High dough yield, low dough consistency, and high bake loss were favourable in oat baking, since they correlated with high specific volume (*p* < 0.001) and soft crumb structure (*p* < 0.01). Bread crumb hardness correlated strongly between the storage days one and three (*p* < 0.001), indicating that the initial crumb hardness mainly determined the crumb hardness during storage. Low dough consistency, high bake loss, and high specific volume correlated with the low staling rate of the bread (*p* < 0.05).

Among the physicochemical quality parameters of the oat flour samples, only the fat content and the particle size of oat flours (D_50_, D_4,3_) correlated with more than one baking quality parameter. High fat content of oat flours correlated with low dough consistency (*p* < 0.05), high specific volume (*p* < 0.01), and high bake loss (*p* < 0.01). Small median particle size (D_50_) correlated with high dough yield and high bake loss (*p* < 0.05), and low crumb hardness (*p* < 0.01). All particle size parameters (D_50_, D_4,3_, D_90_) correlated positively with the staling rate (*p* < 0.05), indicating that the larger particle size was related to the greater staling rate of the bread. WHC and protein, starch, ash, and total dietary fibre contents of the oat flours did not correlate with any of the baking quality parameters (*p* > 0.05). β-glucan content of the oat flours showed only a weak correlation with crumb hardness, and only after one day of storage (*p* < 0.05).

## 4. Discussion

The objective of this study was to assess the baking quality variation of twenty whole grain oat cultivar samples, and to identify the factors that caused the variation. In previously published whole oat baking studies, oat cultivars and processing steps were not controlled [16], or the oat was not properly kilned [17]. Kiln treatment is included in oat processing to increase the shelf-life of oat and to prevent enzymatic rancidity, since oat contains high levels of fat and lipolytic enzymes [18]. In the mixed oat-wheat baking, the insufficient treatment of oat flours has been reported to decrease the bread specific volume and impair the bread crumb texture [12]. In our study, all oat cultivar samples were grown in Finland, and they were processed (kilned and milled) identically at the same mill. We observed that the baking quality varied greatly between the oat cultivar samples, although most of the samples were baked with good quality by dough yield optimisation. Physicochemical variation of the oat cultivar samples played a role in baking quality variation, as part of the physicochemical quality factors significantly correlated with the baking quality parameters.

In our study, most of the oat cultivar samples were baked with good or excellent quality by the dough yield optimisation. Previously, specific volumes of 1.14–1.66 mL/g have been reported for whole oat breads [16,17], while in our study, specific volumes were greater (1.45–1.93 mL/g). However, a good loaf bread shape using wheat baking quality criteria was challenging to obtain with our gluten-free baking method. In commercial gluten-free and non-wheat breads, bread quality is usually improved with several hydrocolloids but in our study, simple recipe with only one hydrocolloid (psyllium) allowed better assessment of the baking quality as fewer components were involved in baking. The loaf bread method clearly demonstrated the variation in baking quality, and allowed better crumb structure evaluation. For comparison, flat yeast-proofed bread, which is a common type of oat and rye breads in Nordic countries (pala bread technology, [19]), enables the use of high amounts of non-wheat flours and fibre components in the dough without impairing the bread quality but in that case, baking quality variation might be challenging to evaluate.

The optimal dough yield (water absorption of the dough) was a sample-specific trait, and too high and too low dough yields decreased the baking quality. However, high dough yield or low dough consistency could not be used alone for optimisation, as all samples could not be baked at high dough yields with good quality. In addition, optimal dough yield could not be predicted from the physicochemical quality factors of the oat cultivar samples, as only the median particle size correlated with optimised dough yield. Test baking is considered a time-consuming optimising method, but there is no other method to optimise oat baking quality, where both dough workability and bread quality are included. In wheat baking, it has been reported that the differences in bread volume became apparent during oven baking and could not be detected in earlier stages of baking [8,20]. In our study, we optimised the water absorptions of the dough separately for each sample, while baking process conditions were similar for all samples. This was in accordance with previous publications, where gluten-free baking has been optimised according to the water content of the dough or the recipe, while proofing and baking conditions have been constant [9,10,21,22].

In our test baking trials, too high dough yields were eliminated as the doughs were too sticky to handle, or they caused impaired bread crumb structure. According to Eliasson & Larsson [8], decreased wheat baking quality at too high dough yields resulted from the insufficient strength of the gas cell interfaces, and thus, the dough structure was not strong enough to retain the excess water. Also, according to Bloksma [23], sufficient gas cell stabilization is needed in the dough so that the gas cell membranes do not rupture prematurely during fermentation or oven rise and thus, a bread of high volume can be obtained. Rapid bread volume expansion during baking due to temperature increase is a critical point for the gas cells [8]. Also, in gluten-free baking, too liquid doughs have been reported to result in breads low in volume as the gas was not retained [10]. In our study, also too low dough yields decreased the baking quality, as the crumb structure remained dense and low in porosity. In gluten-free doughs, too low water additions [10] and too rigid doughs [24] have been related to low specific volumes and high crumb firmness. Thus, as hypothesised, the optimal water absorption of the oat dough was essential for good baking quality.

High optimal dough yield was beneficial in whole grain oat baking, and the advantages of high water content in the dough have been demonstrated also earlier in gluten-free baking [9,10,21]. Possibly in our study, higher dough yields (more water in the dough) allowed more effective mixing of the dough and resulted in more gas cell nuclei for the yeast fermentation and better hydration of the flour components. Air incorporation into the dough during mixing is crucial, as new air cells cannot be formed in the dough during proofing and baking [8]. In our study, the mixing intensities were not increased in the doughs of lower dough yields to compensate for the possible differences in mixing efficiency, since it would have created a new variable in our study, and increased mixing intensity could have caused dough temperature to increase. In our study, higher dough yield led to higher bake loss (i.e., more water evaporation during baking) and allowed a porous and aerated crumb structure to be formed. High bake loss seemed to be a prerequisite for good baking quality, as bake loss correlated strongly with the bread specific volume (r = 0.949, *p* < 0.001). The role of water evaporation during baking in bread structure formation has been discussed also previously [8,23,25].

In our study, high dough yield correlated negatively with bread crumb hardness (on day 3, r = −0.708, *p* < 0.001), and thus, high dough yield was related to softer bread crumbs. Also earlier, high water content in the dough has been reported to decrease bread hardness in gluten-free baking [9,10]. In our study, oat cultivar samples varied greatly regarding the crumb hardness and the staling rates of the breads. In 30% of the oat cultivar samples, the crumb hardness did not increase at all during the storage while in 10% of samples, even an 8 N increase in crumb hardness was observed. It has been discussed that the loaf volume greatly influences the wheat bread hardness [8]. In our study, bread specific volume correlated negatively with bread crumb hardness (on day 3, r = −0.746, *p* < 0.001). Thus, the greatest specific volumes led to the softer crumbs (and consequently, highest crumb porosities), whereas the hardest breads had the lowest porosities and the tightest structures in the crumb. Hence, gas cell stabilization was also related to the oat bread structure and texture.

Among the chemical data of the oat cultivar samples, only the fat content of the oat flour samples correlated with the baking quality parameters; with dough consistency (r = −0.474, *p* < 0.05), bread specific volume (r = 0.629, *p* < 0.01), and bake loss (r = 0.632, *p* < 0.01). Previously, the positive baking properties of oat oil have been demonstrated in wheat baking [26,27]. Later, the foaming properties of polar oat lipids and the foam-active role of tryptophanins, oat lipid-binding proteins, have been reported [28]. The technological role of the oat lipids in oat baking is consistent with the recent findings that in oat dough, dough liquor and air/water interfaces of the gas cells are dominated by lipids whereas, in wheat doughs, the interfaces are co-stabilized by proteins and lipids [29,30]. Our result, that the fat content of the oat flour correlated with the bread specific volume, refers to the role in gas-cell stabilization. The importance of lipids, especially polar lipids, in wheat baking quality, has been discussed also previously [8,20,31].

The other chemical quality factors of the oat flour samples such as protein, starch, total dietary fibre, and ash contents had no correlations with the baking quality parameters. β-glucan content of the oat flours showed only a weak positive correlation with bread crumb hardness, and only after one day of storage (r = 0.492, *p* < 0.05). Previously, the baking quality of gluten-free rice bread increased after β-glucan addition at low concentrations (1–1.3% of the flour weight) while at higher concentrations (2% or more), the baking quality decreased [22,24]. In our baking trials, β-glucan concentrations were considerably higher in the doughs compared to the literature (2.9–4.6% of the flour weight).

Among the physicochemical quality factors of the oat flour samples, only the median particle size (D_50_) correlated with optimised dough yield (r = −0.496, *p* < 0.05). The relation between oat flour particle size and optimal dough yield has not been published before. In gluten-free sorghum baking with optimised water addition, the small particle size of the flours increased the bread specific volume and decreased the bread hardness [32]. In our study, the median particle size of the oat flours correlated with bake loss (r = −0.532, *p* < 0.05), crumb hardness (day 1, r = 0.609, *p* < 0.01; day 3, r = 0.726, *p* < 0.001), and staling rate (r = 0.592, *p* < 0.01). Thus, small particle size was related to the higher water absorption of the dough, higher water evaporation during baking, lower bread hardness, and lower staling rate. Although the oat flour samples of our study were milled in similar conditions, the particle sizes of the samples varied and caused variation in the oat baking quality, as well. However, we could not identify the cause of the particle size variation in this study, and thus, further research is needed to understand better how to control the quality of the whole grain oat baking.

## 5. Conclusions

In our study, it was observed that dough yield optimisation was an important quality factor in whole grain oat baking, as most of the oat cultivar samples were baked with good quality by dough yield optimisation. However, variation was observed in the baking quality of twenty oat cultivar samples, as all samples could not be baked with good quality at high dough yields. In whole grain oat baking at optimised dough yield, high fat content and small median particle size of the oat flours had a significant positive influence on the baking quality. This influence needs to be studied in more detail to understand the baking quality of whole grain oat at a deeper level. However, the results of this study can already help millers and bakers to optimise and develop whole grain oat baking in the industrial level processes.

## Figures and Tables

**Figure 1 foods-10-02461-f001:**
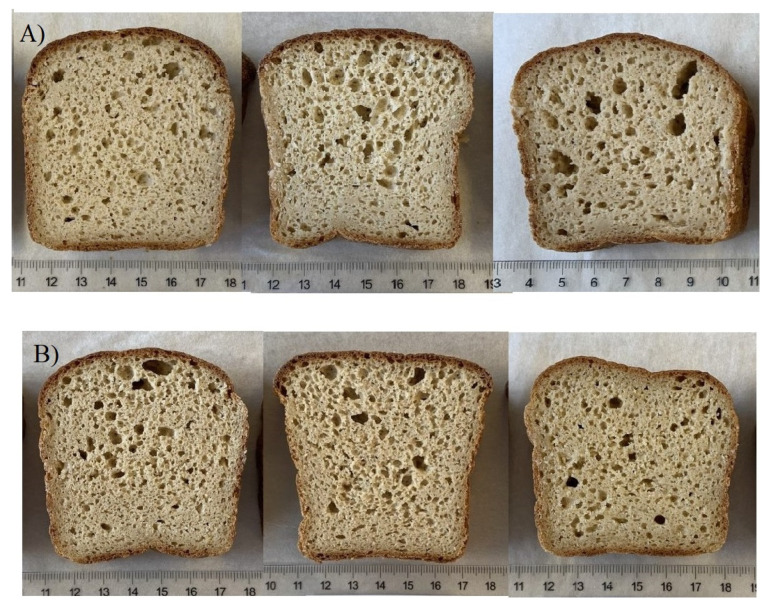
(**A**) Bread crumb structures rated 3 (excellent), 2 (good), and 1 (satisfactory), from left to right. (**B**) Bread shapes rated 3 (excellent), 2 (good), and 1 (satisfactory), from left to right.

**Figure 2 foods-10-02461-f002:**
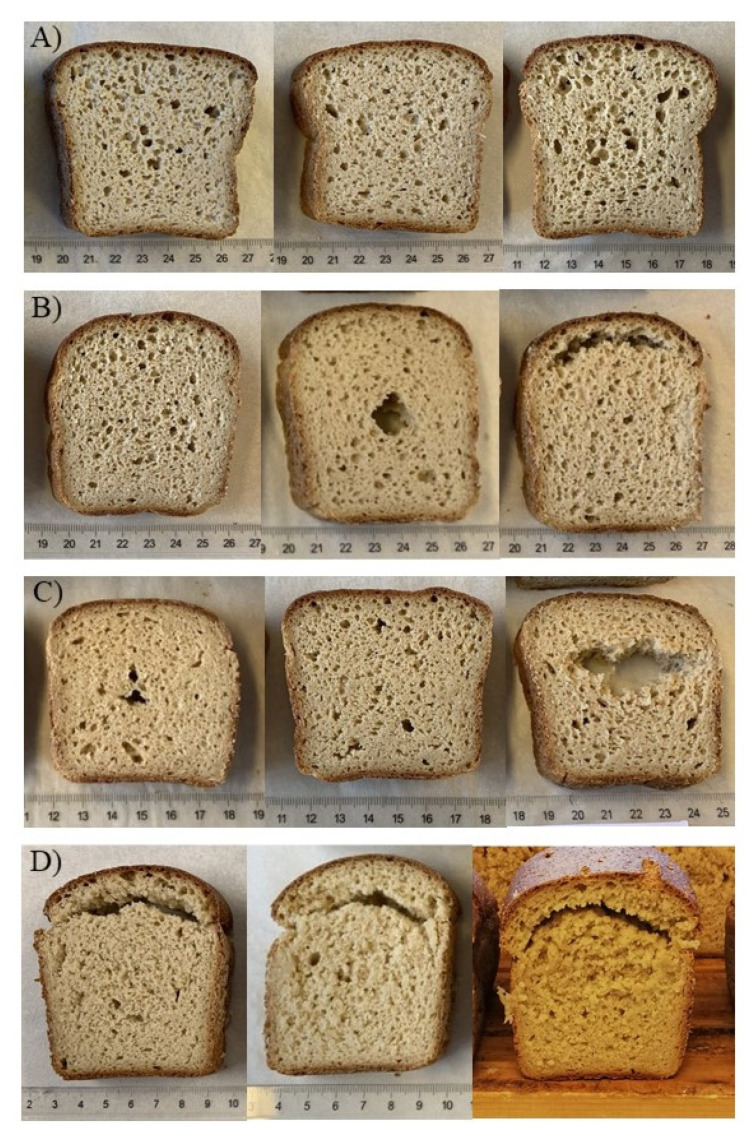
(**A**) The breads from three oat cultivar samples baked at optimal dough yield of 205. (**B**) Oat cultivar sample (F28) baked at dough yields 197, 200, and 205, from left to right. (**C**) Oat cultivar sample (F21) baked at dough yields 185, 192, and 200, from left to right. (**D**) Oat cultivar sample (F23) baked at dough yields 185, 190, and 195, from left to right.

**Figure 3 foods-10-02461-f003:**
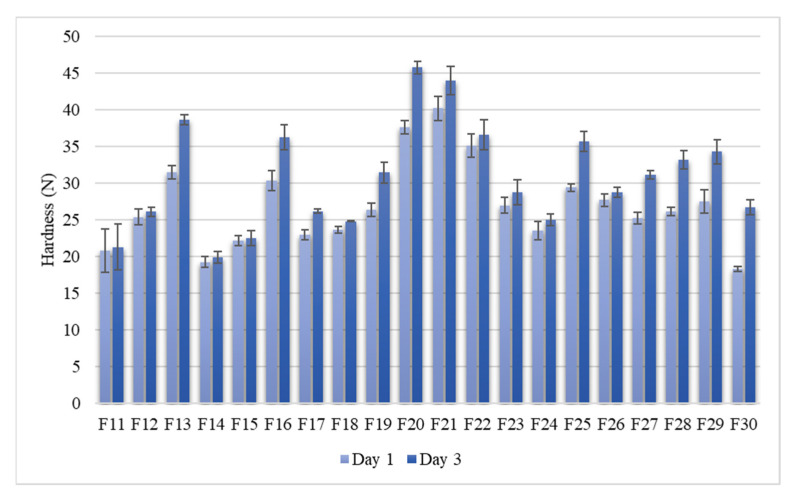
Bread crumb hardness (N) of the oat bread samples after 1 and 3 days of storage (*n* = 3), measured with two-bite compression test of Texture Analyser (TPA, Texture Profile Analysis, Stable Micro Systems, UK). Error bars represent standard errors of the means (SEM).

**Table 1 foods-10-02461-t001:** The particle size distribution parameters (D_50_, D_4,3_, D_90_) and centrifugal WHC of the oat flour samples (*n* = 20) with the maximum, minimum, and average values of three replicate measurements. Error values represent standard errors of the means (SEM).

Sample	D_50_ (µm), *n* = 3	D_4,3_ (µm), *n* = 3	D_90_ (µm), *n* = 3	WHC (mL/g), *n* = 3
F11	119 ± 2	229 ± 3	606 ± 8	1.08 ± 0.00
F12	116 ± 1	239 ± 2	642 ± 5	1.11 ± 0.05
F13	203 ± 6	325 ± 1	823 ± 7	1.32 ± 0.06
F14	112 ± 4	236 ± 4	641 ± 7	1.24 ± 0.05
F15	135 ± 2	252 ± 1	667 ± 4	1.45 ± 0.00
F16	159 ± 4	260 ± 4	668 ± 10	1.34 ± 0.06
F17	158 ± 5	292 ± 4	767 ± 5	1.16 ± 0.00
F18	167 ± 2	300 ± 2	784 ± 8	1.19 ± 0.00
F19	146 ± 1	259 ± 3	671 ± 9	1.21 ± 0.00
F20	170 ± 5	277 ± 3	712 ± 4	1.29 ± 0.00
F21	204 ± 1	316 ± 2	796 ± 7	1.25 ± 0.06
F22	146 ± 3	252 ± 4	656 ± 11	1.19 ± 0.00
F23	143 ± 4	270 ± 5	714 ± 12	1.09 ± 0.06
F24	119 ± 1	257 ± 6	704 ± 23	1.19 ± 0.00
F25	147 ± 4	284 ± 7	761 ± 17	1.19 ± 0.00
F26	154 ± 6	260 ± 6	675 ± 12	1.13 ± 0.00
F27	174 ± 5	314 ± 4	826 ± 9	1.20 ± 0.00
F28	159 ± 1	268 ± 3	691 ± 9	1.25 ± 0.07
F29	192 ± 6	304 ± 4	764 ± 12	1.14 ± 0.00
F30	148 ± 2	278 ± 3	748 ± 13	1.08 ± 0.00
Min−Max	112–204	229–316	606–826	1.08–1.45
Average	153	274	716	1.20

D_50_ median particle diameter. D_4,3_ average particle diameter. D_90_ diameter, that 90% of the particles are smaller. WHC water holding capacity.

**Table 2 foods-10-02461-t002:** Optimised dough yields and dough consistencies, bake losses, specific volumes, and staling rates of the oat bread samples (*n* = 20).

Sample	Optimised Dough Yield ^1^	Dough Consistency (N∙s), *n* = 5	Bake Loss (%), *n* = 3	Specific Volume (mL/g), *n* = 3	Staling Rate (N), *n* = 3
F11	205	135 ± 3 ^i^	17.3 ± 0.3 ^ab^	1.9 ± 0.03 ^ab^	0
F12	195	218 ± 5 ^ef^	16.2 ± 0.2 ^cde^	1.71 ± 0.02 ^c^	0
F13	198	234 ± 5 ^de^	15.4 ± 0.2 ^efg^	1.58 ± 0.02 ^de^	7.1 ± 0.6
F14	200	156 ± 5 ^i^	17.6 ± 0.1 ^a^	1.91 ± 0.02 ^ab^	0
F15	205	162 ± 4 ^hi^	17.3 ± 0.2 ^ab^	1.93 ± 0.01 ^ab^	0
F16	195	292 ± 8 ^b^	15.9 ± 0.2 ^def^	1.63 ± 0.02 ^cde^	5.9 ± 0.7
F17	205	136 ± 4 ^i^	17.4 ± 0.1 ^ab^	1.92 ± 0.01 ^ab^	3.2 ± 0.4
F18	195	194 ± 3 ^fg^	17.0 ± 0.1 ^abc^	1.85 ± 0.01 ^b^	1.1 ± 0.5
F19	198	267 ± 6 ^bc^	15.9 ± 0.1 ^def^	1.57 ± 0.02 ^de^	5.1 ± 0.7
F20	190	359 ± 7 ^a^	14.9 ± 0.1 ^g^	1.45 ± 0.01 ^f^	8.1 ± 1.2
F21	192	255 ± 7 ^cd^	15.6 ± 0.1 ^defg^	1.59 ± 0.02 ^de^	3.8 ± 0.8
F22	190	262 ± 8 ^c^	15.2 ± 0.1 ^fg^	1.56 ± 0.01 ^def^	1.5 ± 0.8
F23	185	215 ± 5 ^ef^	16.4 ± 0.1 ^cd^	1.98 ± 0.05 ^a 2^	1.8 ± 0.9
F24	200	207 ± 3 ^efg^	16.2 ± 0.2 ^cdef^	1.60 ± 0.02 ^de^	0
F25	197	231 ± 7 ^de^	15.6 ± 0.2 ^defg^	1.52 ± 0.01 ^def^	6.0 ± 2
F26	197	192 ± 4 ^fg^	16.7 ± 0.1 ^bcd^	1.84 ± 0.01 ^b^	0
F27	192	230 ± 3 ^de^	15.8 ± 0.2 ^defhg^	1.65 ± 0.01 ^cd^	5.9 ± 0.9
F28	197	225 ± 5 ^e^	16.3 ± 0.2 ^cde^	1.83 ± 0.02 ^b^	7.0 ± 2
F29	192	220 ± 5 ^ef^	15.5 ± 0.1 ^defg^	1.54 ± 0.01 ^def^	7.0 ± 3
F30	197	184 ± 5 ^gh^	15.9 ± 0.2 ^def^	1.57 ± 0.01 ^de^	8.4 ± 1.3
Min−Max	185–205	135–359	14.9–17.6	1.45–1.93	0–8.4
Average	196	219	16.2	1.71	3.6

Error values represent standard errors of means (SEM). Different superscript lowercase letters (a–i) in the same column indicate statistically significant (*p* < 0.05) differences. ^1^ At 14% moisture content of the oat flours. ^2^ The sample F23 with a torn crumb structure, and therefore the specific volume was a result of a hole inside the bread.

**Table 3 foods-10-02461-t003:** Pearson’s correlation coefficients of the baking quality parameters with each other and with the physicochemical quality parameters of the oat cultivar samples (*n* = 19).

	Dough Yield	Dough Consistency	Bake Loss	Specific Volume	Crumb Hardness, Day 1	Crumb Hardness, Day 3	Staling Rate
**Baking Quality Parameter**							
Dough yield	1	−0.765 ***	0.791 ***	0.697 ***	−0.674 **	−0.708 ***	-
Dough consistency		1	−0.845 ***	−0.780 ***	0.786 ***	0.849 ***	0.542 *
Bake loss			1	0.949 ***	−0.731 ***	−0.858 ***	−0.674 **
Specific volume				1	−0.607 **	−0.746 ***	−0.644 **
Crumb hardness, day 1					1	0.925 ***	-
Crumb hardness, day 3						1	0.639 **
Staling rate							1
**Physicochemical Quality Parameter**							
Protein ^a^	-	-	-	-	-	-	-
Fat ^a^	-	−0.474 *	0.632 **	0.629 **	-	-	-
Starch ^a^	-	-	-	-	-	-	-
Ash ^a^	-	-	-	-	-	-	-
TDF ^a^	-	-	-	-	-	-	-
β-glucan ^a^	-	-	-	-	0.492 *	-	-
WHC	-	-	-	-	-	-	-
D_50_	−0.496 *	-	−0.532 *	-	0.609 **	0.726 ***	0.592 **
D_4,3_	-	-	-	-	-	0.530 *	0.555 *
D_90_	-	-	-	-	-	-	0.516 *

Significance levels: * *p* < 0.05, ** *p* < 0.01, *** *p* < 0.001, - not significant (*p* > 0.05). ^a^ Published previously by [1]. Reprinted with permission from ref [1]. Copyright 2021 Jokinen et al. TDF total dietary fibre. WHC water holding capacity. D_50_ median particle diameter. D_4,3_ average particle diameter. D_90_ diameter, that 90% of the particles are smaller.

## Data Availability

The data presented in this study are available on request from the corresponding author. The data are not publicly available due to the agreement of the project.

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
