# Peer review of "Baking Quality Assessment of Twenty Whole Grain Oat Cultivar Samples"

_foods, 2021, doi:10.3390/foods10102461_

Round 1

Reviewer 1 Report

please add numerical data to the abstract to be more concrete

page 2, lines 49-51: From the point of view of baking value this is true, but the statement "gluten-free" may mislead some readers, because although oats are generally considered safe for people intolerant to gluten, some varieties of oats can be as harmful to such people as, for example, wheat (see e.g. World J Gastroenterol. 2015 Nov 7; 21(41): 11825–11831 doi: 10.3748/wjg.v21.i41.11825; Nutrients. 2019 Oct; 11(10): 2345 https://doi.org/10.3390/nu11102345). Please take this fact into account and rewrite this sentence.

page 3, line 114: please specify what is a light syrup

page 8, lines 241-242: This is surprising as an increase in crumb hardness is one of the main symptoms of aging in all types of bread, can the Authors give any explanation for their observations?

page 10, lines 306-309: to be precise, hydrocolloids have been also used in your study, however not in pure form, but as psyllium seed

page 11, line 348: What changes take place in the dough, what exactly the effect of better mixing is due to? Would it not be possible to achieve this effect with a smaller addition of water by using more intensive mixers?

Author Response

Dear Reviewer,

We are grateful for your review. Please see the PDF attachment for our point-by-point responses to your comments, and our edited manuscript (TrackChanges version).

Sincerely,

Saara Sammalisto (corresponding author)

Reviewer 2 Report

In the presented study the authors detailly examined numerous cultivars of oats in terms of their suitability to produce gluten-free breads of high quality.

The paper represents a valuable scientific work with a large application potential.

Only a few fragments of the text require a minor corrections, namely:

  • line 49-51 - Oat products can be considered gluten-free not because of the lack of viscoelestic network, but because of the lack of toxic sequences in protein structure.
  • line 120 - "moulding" instead of "baking"
  • line 177 - "to" instead of "than in"
  • lines 279, 384 - "β-glucan" instead of "β-Glucan" 
  • line 359-360 - begin the sentence with "Also earlier"

Author Response

Dear Reviewer,

Thank you for your review. Please find attached our point-by-point responses to your comments.

Sincerely,

Saara Sammalisto (corresponding author)
